# Sleeping at the switch

Maude Bouchard[1,2]*, Jean-Marc Lina[1,3,4], Pierre-Olivier Gaudreault[1], Alexandre Lafrenière[1], Jonathan Dubé[1,2], Nadia Gosselin[1,2], Julie Carrier[1,2]*

[1]Center for Advanced Research in Sleep Medicine, Hôpital du Sacré-Coeur de Montréal, Montreal, Canada; [2]Department of Psychology, Université de Montréal, Montreal, Canada; [3]Department of Electrical Engineering, École de Technologie Supérieure, Montreal, Canada; [4]Centre de Recherches Mathématiques, Université de Montréal, Montreal, Canada

**Abstract** Sleep slow waves are studied for their role in brain plasticity, homeostatic regulation, and their changes during aging. Here, we address the possibility that two types of slow waves co-exist in humans. Thirty young and 29 older adults underwent a night of polysomnographic recordings. Using the *transition frequency,* slow waves with a slow transition (slow switchers) and those with a fast transition (fast switchers) were discovered. Slow switchers had a high electroencephalography (EEG) connectivity along their depolarization transition while fast switchers had a lower connectivity dynamics and dissipated faster during the night. Aging was associated with lower temporal dissipation of sleep pressure in slow and fast switchers and lower EEG connectivity at the microscale of the oscillations, suggesting a decreased flexibility in the connectivity network of older individuals. Our findings show that two different types of slow waves with possible distinct underlying functions coexist in the slow wave spectrum.

**\*For correspondence:**
maude.bouchard.1@umontreal.ca (MB);
julie.carrier.1@umontreal.ca (JC)

**Competing interests:** The authors declare that no competing interests exist.

## Introduction

Sleep slow waves are non-rapid eye movement (NREM) sleep oscillations in the delta range (<4 Hz) reflecting a high neuronal synchronization (*Diekelmann and Born, 2010*). They generate sustained interest in neuroscience research for their role in sleep-dependent memory consolidation, synaptic plasticity, and as markers of homeostatic sleep pressure (*Gais et al., 2002*; *Inostroza and Born, 2013*; *Steriade, 2006*; *Borbély, 2001*; *Diekelmann and Born, 2010*). In human sleep studies, there is, however, a fundamental question as to whether EEG waves showing oscillations <4 Hz are a unique entity or rather hide two types of slow waves with specific functional roles. In humans, slow EEG frequencies are often divided into two components, slow waves (typically 1–4 Hz) vs slow oscillations (<1 Hz) (*Achermann and Borbély, 1997*; *Mölle et al., 2002*; *Muehlroth et al., 2019*). It has been hypothesized that these two components have different functional molecular regulation mechanisms (*Lee et al., 2004*) and responses to homeostatic pressure (*Achermann and Borbély, 1997*; *Campbell et al., 2006*). More recent studies in mice and humans showed that faster delta frequencies (2.5–4.5 Hz) react differently to sleep deprivation than lower delta frequencies (0.75–2 Hz), suggesting distinct neurophysiological substrates. Precisely, compared to lower delta frequencies, faster delta frequencies showed an increased incidence and power after a sleep deprivation protocol (*Hubbard et al., 2020*). In recent years, *Siclari et al., 2014* showed that delta oscillations (1–4.5 Hz) with distinct cortical origins and distributions are sustained by different synchronization processes. Their team further identified two types of slow waves, widespread, steep (type I) and smaller, more circumscribed (type II) slow waves with only the second type showing homeostatic regulation (*Bernardi et al., 2018*). More recently, animal and human studies brought to light new evidence of two types of slow waves based on the positive and negative state duration of slow waves: one showing a positive correlation and another showing a negative correlation between the two (*Nghiem et al., 2020*). Here, we propose to describe the dichotomy in the delta frequency range

based on a new parameter characterizing the time delay from the maximum negative point of the EEG slow wave to the maximum positive point: the *transition frequency*. Using this parameter, we show two types of slow waves driven by different pressures of homeostatic dissipation and endowed by specific EEG functional connectivity dynamics.

Sleep slow waves are characterized by a hyperpolarizing state (a negative phase in surface EEG/ a down-state in animal literature) during which cortical neurons are synchronously silent, followed by a depolarizing state (a positive phase in surface EEG/an up-state in animal literature) during which cortical neurons fire intensively (*Csercsa et al., 2010*; *Steriade, 2006*; *Chauvette et al., 2011*). The transition from the negative to the positive phase is critical, as it is a strong marker of the ability of brain networks to efficiently switch from a state of hyperpolarization to a state of massive depolarization. The slope of the slow wave (the rate of amplitude change from the negative to the positive peak) is associated with the recruitment/decruitment of the neuronal population, with a steeper slope showing a quicker recruitment (*Vyazovskiy et al., 2011*). It is generally described as the best measure to assess the synaptic strength and sleep homeostasis compared to other classic parameters (*Bersagliere and Achermann, 2010*; *Riedner et al., 2007*). However, using the slope as a measure of transition speed also presents important limitations, as it is affected by slow wave amplitude: with similar positive and negative durations, higher slow waves will necessarily have steeper slopes (*Bersagliere and Achermann, 2010*). A novel metric that captures the transition speed and that is more independent of amplitude needs to be developed.

The study of slow waves necessarily involves the notion of age, as slow waves drastically change during adulthood. Compared to young adults, older individuals show lower slow wave density as well as reduced amplitude, smoother slope, and longer positive and negative phase durations of slow waves, possibly indicating that cortical neurons enter less synchronously into the hyperpolarization and depolarization phases (*Carrier et al., 2011*). Therefore, our ability to disentangle the influence of the slow wave amplitude over the slope is compromised in aging. A novel metric that captures the transition speed without being affected by amplitude needs to be developed, especially when studying older populations. Age-related EEG connectivity modification has also been recently described in the literature (*Ujma et al., 2019*; *Bouchard et al., 2019*) with major age-related differences in deeper NREM sleep, when slow waves are prominent. At the scale of sleep stages and cycles, our team reported that older individuals showed higher between-region EEG connectivity at the whole-brain scale in deep NREM sleep (stage N3) as compared to younger adults (*Bouchard et al., 2019*). These results support the notion that the brain of younger individuals during deeper NREM sleep stages operates with reduced long-range cortico-cortical connectivity (*Spoormaker et al., 2011*; *Massimini et al., 2007*). However, EEG connectivity at the scale of slow waves has yet to be studied in humans and during aging. Such information would allow a better understanding of the dynamic and distinct networks recruited during those oscillations in addition to providing functional clues to support the complementary phenomena happening in the delta frequency range as described in other studies. The goals of our study were thus to clearly identify the dichotomy in the slow wave's spectrum and describe the EEG connectivity patterns and homeostatic decline of these two types of slow waves in young and older individuals.

## Materials and methods

### Participants and protocol

Fifty-nine participants, 30 young (14 women, 16 men; 20–30 years; mean = 23.49 ± 2.79 yo) and 29 older (18 women, 11 men; 50–70 years; mean = 59.6 ± 5.61 yo) adults in good physical and mental health, have completed the study protocol (demographic data for each group is presented in *Supplementary file 1*). Exclusion criteria were first investigated during a phone screening using a semi-structured interview. Smoking, a body mass index (BMI) over 27, the use of drugs and/or medication that could affect the sleep-wake cycle and/or the nervous system, complaints about the sleep-wake cycle and/or cognition, transmeridian travel within 3 months prior to the study, and night-shift work or night-shift work in the last 3 months all resulted in the exclusion of the participant. Participants included in the study were asked to maintain between 7 and 9 hr of sleep per night prior to the study. Participants with a score higher than 13 at the Beck Depression Inventory (*Beck et al., 1988a*) or a score higher than 7 at the Beck Anxiety Inventory (*Beck et al., 1988b*) were

excluded from the study. Potential cognitive impairment and dementia were screened using a neuropsychological assessment in which intelligence quotient (IQ), memory, attention, processing speed, and executive functions were performed and ruled out for all participants. Premenopausal women had regular menstrual cycles (25–32 days), and menopausal women showed amenorrhea, for at least a year before the testing. They reported no night sweats or hot flashes. Perimenopausal women were excluded from the research. The protocol was approved by the ethics committee of the Hôpital du Sacré-Coeur de Montréal and was performed in accordance with the relevant guidelines and regulations (CMER-RNQ 08-08-002). Participants provided informed consent and received financial compensation for their participation.

## Procedures

All participants underwent one screening and one experimental night of polysomnographic (PSG) recording at the Center for Advanced Research in Sleep Medicine at the Hôpital du Sacré-Coeur de Montréal. For the screening night only, PSG also included leg electromyogram (EMG), thoracoabdominal plethysmograph, oral/nasal canula as well as frontal, central, and parietal electrodes referred to linked earlobes (*Iber et al., 2007*). Participants with periodic leg movements or sleep apneas/hypopneas (index >10 per hr of sleep associated with a microarousal) were excluded from the study.

## Polysomnographic recording for the experimental night

All participants filled out a sleep diary and followed a regular sleep-wake cycle for 7 days before the experimental night based on their individual habitual bedtimes and wake times (± 30 min). Bedtimes and wake times in the laboratory were also based on their own sleep schedules. On the experimental PSG night, 20 EEG derivations (Fp1, Fp2, Fz, F3, F4, F7, F8, Cz, C3, C4, Pz, P3, P4, Oz, O1, O2, T3, T4, T5, T6) referred to linked earlobes were recorded (10–20 international system; EEG: gain 10,000; band-pass 0.3–100 Hz; −6 dB), in addition to chin EMG, electrooculogram (EOG), and electrocardiogram (ECG). Signals were recorded using an amplifier system (grass model 15A54; Natus Neurology, Warwick, Rhode Island, USA) and digitized at a sampling rate of 256 Hz using a commercial software (Harmonie, Stellate Systems, Montreal, Quebec, Canada). Sleep stages (N1, N2, N3, and REM) were visually scored by an electrophysiology technician in 30 s epochs and according to the standard criteria of the American Academy of Sleep Medicine (AASM) (*Iber et al., 2007*), and sleep cycles were identified. Artifacts were first automatically detected (*Brunner et al., 1996*) and then visually inspected by a trained technician. PSG variables for each group for the experimental night are presented in *Supplementary file 1*.

## Slow wave detection

Slow waves were detected automatically on artifact-free NREM (N2 and N3) epochs on all electrodes using previously published criteria (*Dang-Vu et al., 2008*; *Dubé et al., 2015*). Specifically, data was initially filtered between 0.3 and 4.0 Hz using a band-pass filter (- 3 dB at 0.3 and 4.0 Hz; −23 dB at 0.1 and 4.2 Hz), and slow waves were defined according to the following parameters: a negative peak below −40 μV, a peak-to-peak amplitude above 75 uV, the duration of negative deflection between 1500 and 125 ms, and the duration of positive deflection not exceeding 1000 ms.

## Sleep spindle detection

Spindles were automatically detected on artifact-free NREM (N2 and N3) epochs on all electrodes using a previously published algorithm (*Gaudreault et al., 2018*; *Lafortune et al., 2014*; *Martin et al., 2013*). Specifically, the EEG signal was band-pass filtered between 10 and 16 Hz using a linear-phase finite impulse response filter (–3 dB at 10 and 16 Hz). The envelope amplitude of the Hilbert transform of this band-limited signal was smoothed and a threshold was set at the 75th percentile. All events of duration between 0.5 and 3 s were then selected as a spindle. The overlap of a spindle oscillation with a slow wave, characterized by the onset of the spindle between $-\pi$ and $\pi/2$ on the slow wave phase, was defined as a co-occurrence (see *Figure 1A*).

## Slow wave characteristics

For each slow wave, we derived the map between the time and the phase obtained from the Hilbert transform of the filtered slow wave in the delta band ($0.16 - 4 \text{Hz}$). All slow waves were equally time

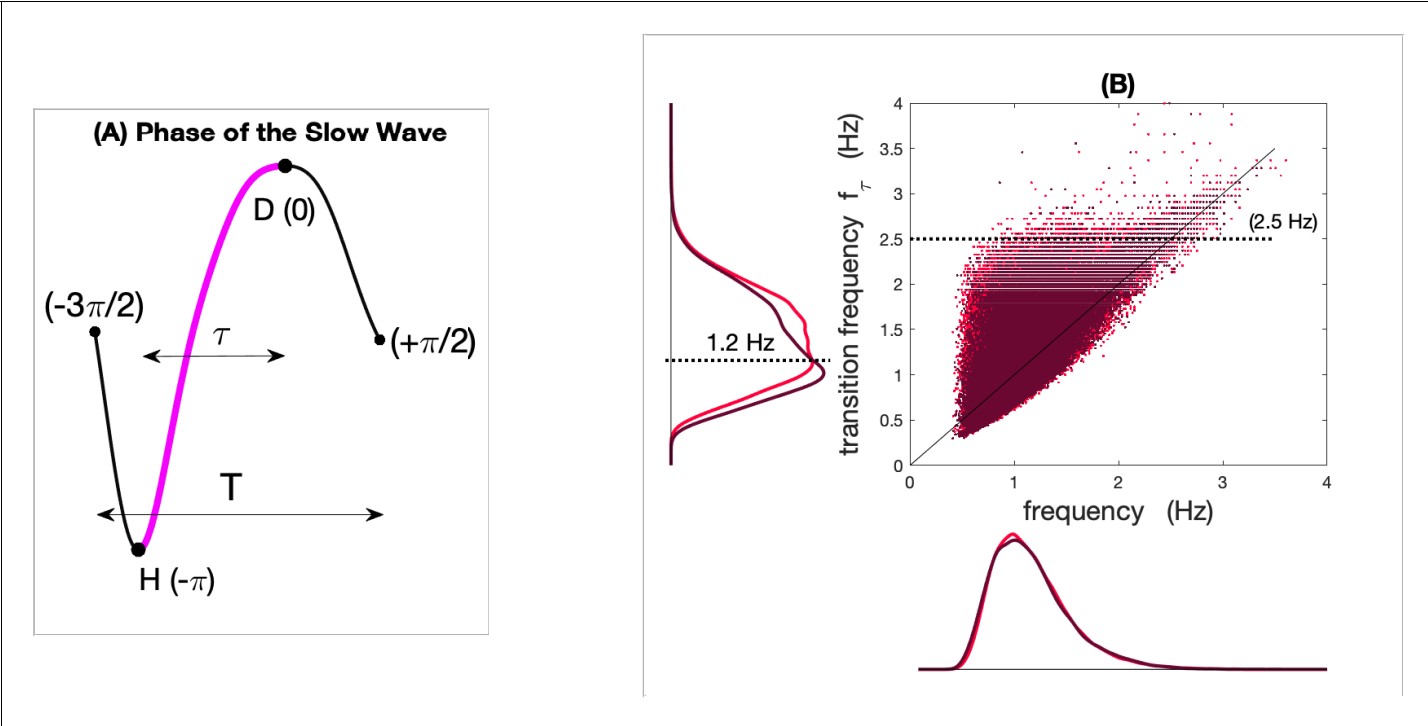

**Figure 1.** Frequency and transition frequency of slow waves in young and older individuals. Panel (**A**) represents the phase of a slow wave with the transition between the maximum negative phase (hyperpolarization (H)) and the maximum positive phase (depolarization (D)) in pink. Panel (**B**) represents a scatter plot of the exhaustive Fz inventory of the frequency ($\frac{1}{T}$) and transition frequency ($\frac{1}{2\tau}$) of each slow wave detected in N2 and N3 in young (light red) and older groups (dark red). The marginal distributions of the two frequencies show a similar distribution for the mean frequency, whereas the transition frequency shows distinct distributions with aging.

referenced by choosing the zero phase at the maximum of the depolarization. Then, the temporal evolution during each slow wave was uniquely described with a phase ranging from $-3\pi/2$ to $\pi/2$ as illustrated in *Figure 1A*. In addition to general parameters like slow wave density (number per min) and frequency (inverse of the total duration $T$), we calculated the *transition frequency* extracted from the filtered slow wave in the delta band. For each slow wave, the transition frequency characterizes the half-wave associated with the depolarization transition. If $\tau$ denotes the delay of the transition from the maximum negative point to the maximum positive point of the slow wave (see *Figure 1A*), then the transition frequency is defined as $f_\tau = 1/2\tau$. *Figure 1B* displays the scatter plot of the overall joint distribution of slow wave frequencies and transition frequencies for all slow waves detected on Fz in young and older individuals. The marginal distributions of the two frequencies clearly show an age difference in the distribution of the transition frequency $f_\tau$ that is not observed for the frequency. We observed a critical value for the *transition frequency* around 1.2 Hz (dashed line) where the two distributions cross with aging. This change in the distribution suggests a model of mixture to reveal distinct modes that could be associated with different types of sleep slow waves that may evolve distinctively with aging.

## Slow and fast switchers

As introduced in the previous section, we considered a mixture of Gaussians to modelize the distribution of the *transition frequency* of the slow wave. As seen in *Figure 2A and B*, distributions show two modes and any slow wave can then be labeled as *slow switchers* or *fast switchers* (cyan and dark blue distribution for young and older participants, respectively). More specifically, the probability distribution can be expanded as a sum of weighted Gaussians $p(f_\tau|Sw)$,

$$p(f_\tau) = p(SlowSw)p(f_\tau|SlowSw) + p(FastSw)p(f_\tau|FastSw)$$

where

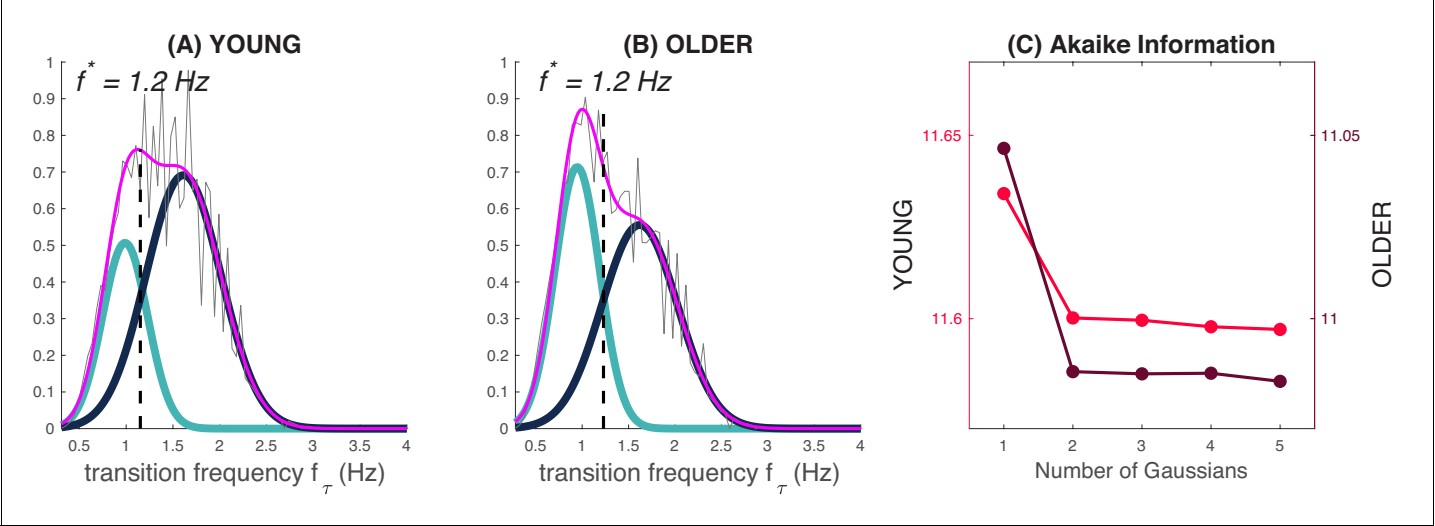

**Figure 2.** Transition frequency and the slow and fast switcher definition. Panels (**A**) and (**B**) are normalized histograms of the transition frequencies (in pink) for young (**A**) and older subjects (**B**), for slow waves detected in N2 and N3 frontal derivations (average of F3, F4, and Fz). The fit of the distributions can be decomposed as a mixture of Gaussians: one Gaussian (cyan) accounts for slow switchers, and the other (dark blue) accounts for the fast switchers. The separation line at f* = 1,2 Hz stands for the intersection between the two Gaussian distributions. Panel (**C**) shows the Akaike Information Criterion for the increasing number of Gaussians in the mixture. The lower the criterion with a sparse decomposition, the better the mixture fit (parietal and central derivations are presented in *Figure 2—figure supplement 2*).

The online version of this article includes the following figure supplement(s) for figure 2:

**Figure supplement 1.** The transition frequency and its relationship with the slope and amplitude of slow waves.

**Figure supplement 2.** Distribution of the transition frequency in central and parietal derivations.

$$p(SlowSw) + p(FastSw) = 1$$

In this sum, $p(f_\tau|SlowSw)$ and $p(f_\tau|FastSw)$ are Gaussian distributions that describe, depending on the class 'slow switchers' or 'fast switchers', the probability to transit with the frequency $f_\tau$. $p(SlowSw)$ (resp.$p(FastSw)$) is the probability for the sleep slow wave to be a slow switcher (resp. a fast switcher). The reliability of the mixture model was further tested with the Akaike Information Criterion (*Figure 2C*) that assessed that a mixture with two Gaussian distributions is necessary and sufficient to fit the entire distribution of the transition frequency. This parametric model of $p(f_\tau)$ can be estimated using the EM (Expectation-Maximization) algorithm to fit the distribution for each individual. From this mixture of Gaussians, we can define the frequency $f^*$ where the two Gaussians intersect: a slow wave will be labeled as a 'slow switcher' if $f_\tau < f^*$, i.e. if $p(f_\tau|SlowSw) \geq p(f_\tau|FastSw)$ and a fast switcher otherwise.

## Slow and fast switcher modulation analysis

To evaluate the decline of slow and fast switchers throughout the night, we calculated the percentage of slow or fast switchers in each sleep cycle related to the respective total number of slow or fast switchers across the night. To statistically test the changes between the slow and fast switchers' decline across sleep cycles, a three-way analysis of variance (ANOVA) with one factor [2 (Group: younger vs older)] and two repeated measures [2 (Switcher: slow vs fast)] × [3 (Cycle: cycle 1, 2, and 3)] was performed. p-values <0.05 were considered significant and simple effects were analyzed to follow up significant interactions.

## Phase-locked connectivity analyses

The functional connectivity across the EEG derivations was assessed using a *time-resolved* phase lag index (PLI) calculated at six successive phases of the slow wave. Five phases were evenly spaced during the transition, whereas a sixth phase was defined after the depolarization maxima. Given a slow

wave (further labeled by $k$) was detected on the derivation denoted by $n^*$, we considered the internal phase of the detected oscillation and the simultaneous phase of the other EEG derivations, $\varphi_{n^*}^{(k)}$ and $\varphi_m^{(k)}$, respectively (the * indicates the derivation on which the slow wave was detected). The slow wave PLI between $n^*$ and any other derivation $m$ is then defined by

$$pli(n^*, m) = \frac{1}{N^*} \sum_{kk} sign\left(sin\left(\varphi_{n^*}^{(k)} - \varphi_m^{(k)}\right)\right)$$

where the summation runs over the $N^*$ slow waves detected on $n^*$. This quantity is calculated for six regularly spaced phases of the detected slow wave. Since this pairwise *pli* emphasizes the slow wave detected on $n^*$, we further symmetrize the definition to account for all the slow waves detected over any pair (n, m):

$$PLI(n, m) = \frac{1}{2}\left(pli(n^*, m) + pli(n, m^*)\right)$$

It is worth noting that slow waves that would truly propagate from $n$ to $m$ with a non-vanishing delay would contribute with $pli(n^*, m) \simeq pli(n, m^*)$ since the non-vanishing delay will be of an opposite sign. The definition of this PLI thus emphasizes the connectivity due to cortico-cortical propagation of the slow waves, independently from the shape of the oscillation. For each of the six phases chosen along the slow waves, the statistical significance of the connections was assessed through a random resampling of the phase with a max-statistics over the full set of electrode pairs. This null-hypothesis modeling was used to define the threshold at each of the six-phase points, for a given p-value (0.01). Finally, to quantify the global strength of the connectivity, a *global connectivity index* (*Bouchard et al., 2019*) was then defined at each of the six slow wave phases as the sum of the PLI over the significant pairs of electrodes. An increasing value of this index qualitatively assesses a more interconnected network or a more significant phase-locked synchronization (with constant non-vanishing delay) among the EEG electrodes.

## Results

### Slow and fast switchers in the sleep slow wave inventory for young and older individuals

*Figure 3* illustrates the slow wave density, distributions of transition frequencies, and the slow switcher probability for frontal (average of F3, F4, and Fz), central (average of C3, C4, and Cz), and parietal (average of P3, P4, and Pz) derivations. As expected, the slow wave density was significantly lower in older individuals as compared to younger participants for the three derivation clusters (*Figure 3A*). Our analyses demonstrated the existence of the '*slow switchers*' and the '*fast switchers*' represented by a bimodal distribution of the transition frequency, in each cluster and for both young and older individuals (*Figure 3B and C*). The cut-off frequency between the two Gaussian curves was statistically determined for each subject and then averaged over both groups (see 'Slow and fast switchers' in the 'Materials and methods' section). In young individuals, a cut-off frequency of 1.3 Hz was found for all derivations, whereas older individuals showed a frequency of 1.2 Hz in frontal and 1.1 Hz in central and parietal derivations. Since further analyses showed that a slow and fast switcher dichotomy exists with or without the concomitant occurrence of a spindle (see *Figure 3—figure supplement 1*), we present the analyses for all slow waves in *Figure 3A,B,C and D*. We also showed that older individuals had a higher probability of producing slow switchers than fast switchers when generating a slow wave compared to younger individuals (*Figure 3D*; p<0.0001 for all derivations). *Figure 3—figure supplement 1* also shows the probabilities to make a slow switcher or a fast switcher slow wave depending on the presence of a co-occurring sleep spindle.

### Fast switchers show a steeper decline than slow switchers at the beginning of night

Considering slow and fast switchers separately, *Figure 4* displays the percentage of switchers in each cycle related to the total number of the same switchers across the whole night. Remarkably, the first three sleep cycles, which are present in almost all participants (cycles 4 and 5 mostly missing

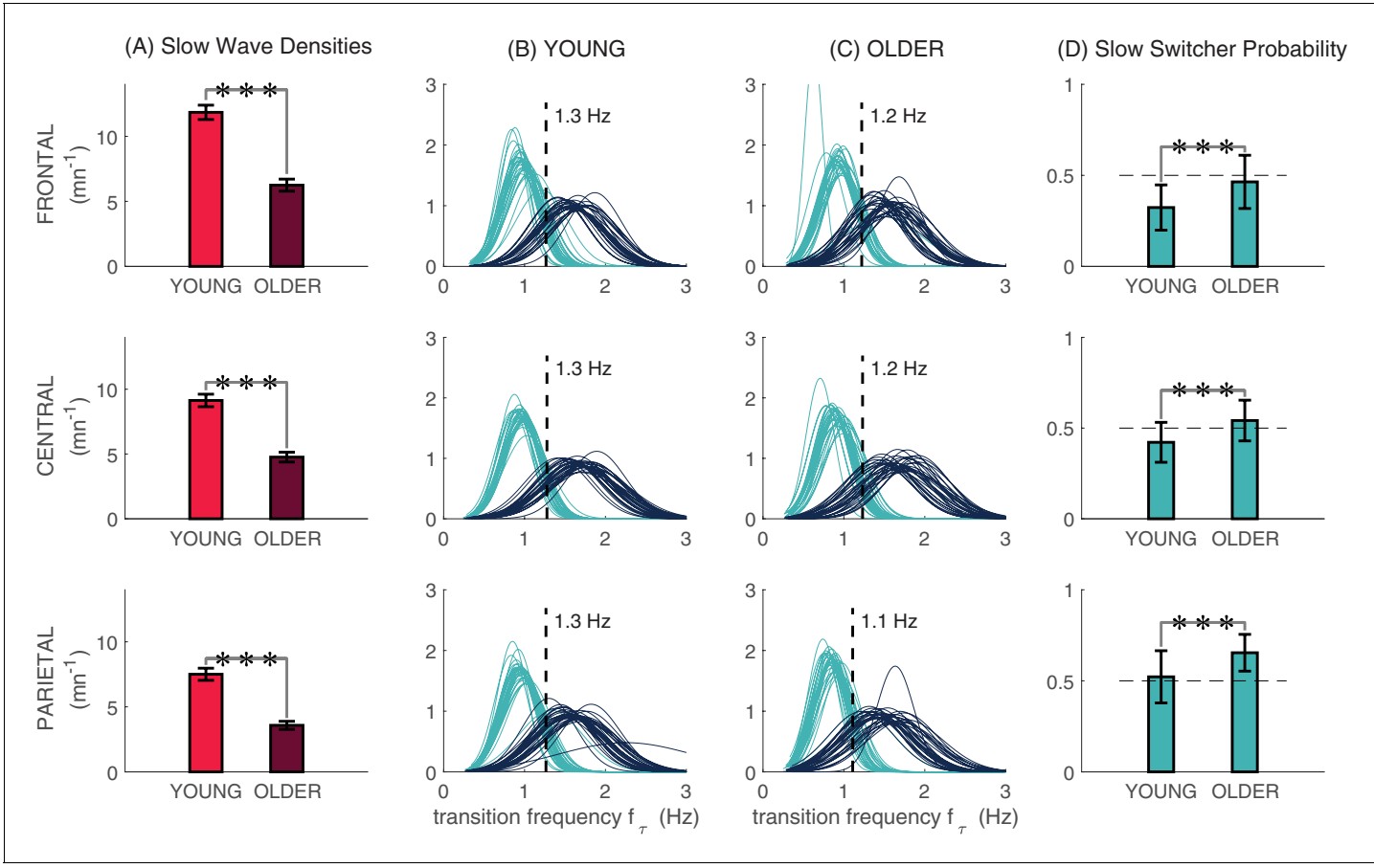

**Figure 3.** Slow and fast switcher slow waves. (A) Slow wave densities in clusters of frontal (F3, Fz, F4), central (C3, Cz, C4), and parietal (P3, Pz, P4) derivations in young and older individuals. Slow waves were detected in N2 and N3. (B and C) The distribution of probabilities of slow waves being slow (cyan) or fast switchers (dark blue) in younger and older individuals, respectively, with each curve representing one participant. We can observe the two distinct modes of sleep slow waves based on their transition frequency in each derivation. (D) Age-related differences in the probability of producing a slow switcher when generating a slow wave. Significant age differences in (A) and (D) were calculated using t-tests (***p<0.0001). The online version of this article includes the following figure supplement(s) for figure 3:

**Figure supplement 1.** Slow and fast switcher slow waves with and without sleep spindles.

for the older individuals), showed an exponential decay that is significantly different for the two types of slow waves. In the younger group, the exponential $\sim e^{-rt}$ drives the fast and slow switchers' decay with $r = 1.6 (R^2 = 1)$ and $r = 1.3 (R^2 = 1)$, respectively. For older subjects, a much slower exponential decay also drives the fast switchers with $r = 0.6 (R^2 = 1)$ whereas the slow switchers are rather evolving with an exponential *reduction* $\sim -e^{rt}$ with $r = 0.4 (R^2 = 1)$.

The three-way ANOVA with repeated measures showed significant Group × Cycle [F(2,1) = 6.9, p = 0.001] and Cycle × Switcher [F(2,1) = 69.2, p<0.001] interactions as well as a specific Cycle effect [F(1.4, 80.1) = 69.7, p<0.001]. Simple effects analysis for the Group × Cycle interaction showed that younger individuals had, in general, more slow waves (averaged number of both types) in Cycle 1 (t (57) = 2.6, p<0.05) but less in Cycle 2 (t(57) = −2.2, p<0.05) when compared to older individuals, whereas no group differences were found for Cycle 3. These results highlight a stronger decrease of slow waves in young individuals as compared to the older individuals between Cycle 1 and Cycle 2, suggesting a steeper decline of homeostatic pressure. As for the Cycle × Switcher interaction, simple effects analysis showed a higher proportion of fast switchers than slow switchers for Cycle 1 (t (58) = −9.6, p<0.001), whereas an opposite effect was found for Cycle 2 (t(58) = 7.0, p<0.001) and Cycle 3 (t(57) = 6.0, p<0.001). When put together, these results suggest a steeper decline of fast switchers between Cycle 1 and Cycle 2 when compared to slow switchers, as demonstrated by the inversion of the slopes at the second time point.

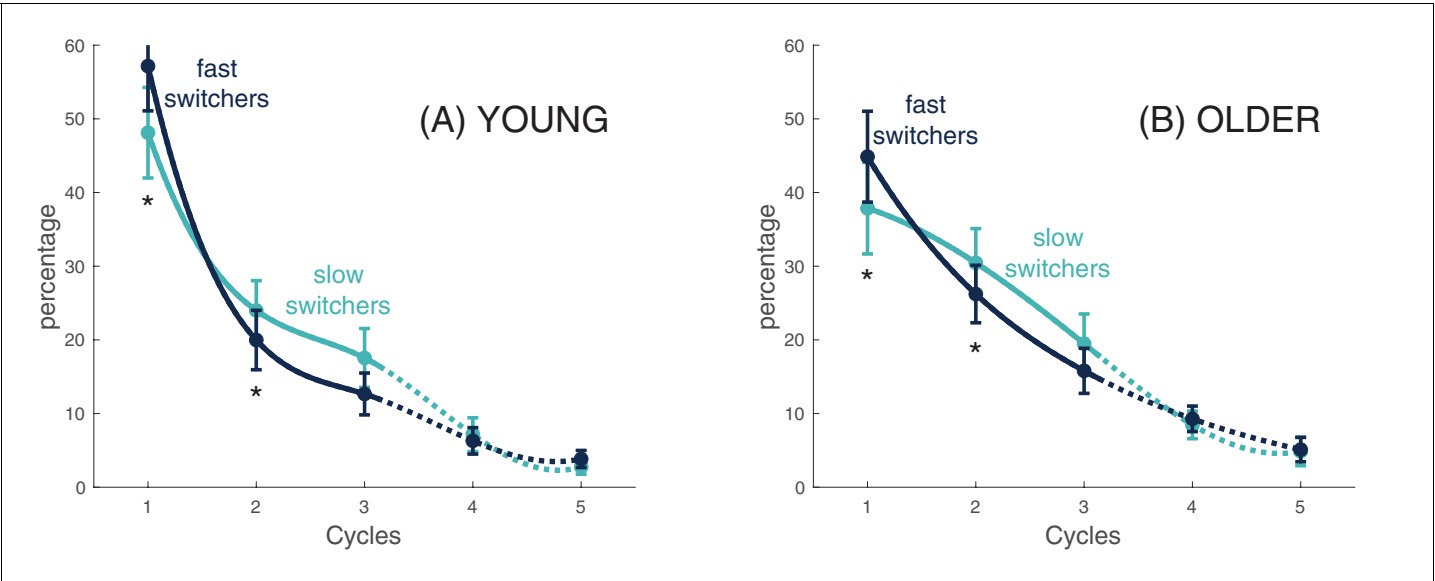

**Figure 4.** Slow and fast switchers decline across sleep cycles. For each participant in each group (Young individuals in Panel A and older individuals in Panel B), we clustered slow switchers and fast switchers with respect to the cycle during which they occurred. Considering slow and fast switchers separately, the percentage of switchers in each cycle related to the total number of switchers across the whole night is displayed. The curves are spline-interpolation of the overall average in each group.

## EEG connectivity dynamics evolve differently within slow and fast switchers

PLI was used to investigate intrinsic slow wave EEG connectivity networks across different phases of the depolarization transition (*Figure 5A*). Since sleep spindles can involve dynamic changes in connectivity (*Zerouali et al., 2014*), analyses were performed separately for spindle-free slow waves and slow waves coupled to a spindle. *Figure 5B and C* show the connectivity graphs across consecutive phases in slow and fast switchers for younger individuals (*Figure 5B*) and older individuals (*Figure 5C*) for slow waves without sleep spindles. A similar analysis for slow waves with sleep spindles of young and older individuals is shown in *Figure 6*. Analysis of the global connectivity index, which quantifies the overall significance of the global connectivity of a graph, showed a distinct EEG connectivity strength for slow and fast switchers in both young (*Figure 5D*) and older (*Figure 5E*) adults. Our results also showed a higher overall EEG connectivity during slow switchers as compared to fast switchers and a higher connectivity in young individuals rather than in older individuals. More specifically, in younger individuals, we observed a global increase in EEG connectivity during slow switchers, which reached its highest connectivity strength at the maximum of depolarization. Whereas this scenario recruited connectivity patterns along the full slow wave depolarization in the absence of a spindle (*Figure 5D*), the connectivity involved in the slow wave with a spindle was concentrated later, around the maximum depolarization phase (*Figure 6D*). In older individuals, slow switchers showed a higher EEG connectivity along the depolarization transition while no significant link was found for the fast switchers. With aging, the fast switchers or the presence of a spindle drastically obliterated the EEG connectivity (*Figures 5E* and *6E*).

## Discussion

In the present work, we have identified two types of slow waves: the *slow switchers* and the *fast switchers*. Slow and fast switchers showed distinct distributions of their *transition frequency* (the transition between the maximum negative point and the maximum positive point of the slow wave) and were detected in both age groups with the older participants showing a higher proportion of slow switchers. We demonstrated that slow and fast switchers are characterized by a specific EEG connectivity signature along the depolarization transition, with slow switchers presenting an overall higher EEG connectivity than fast switchers. Connectivity across slow waves was lower in older

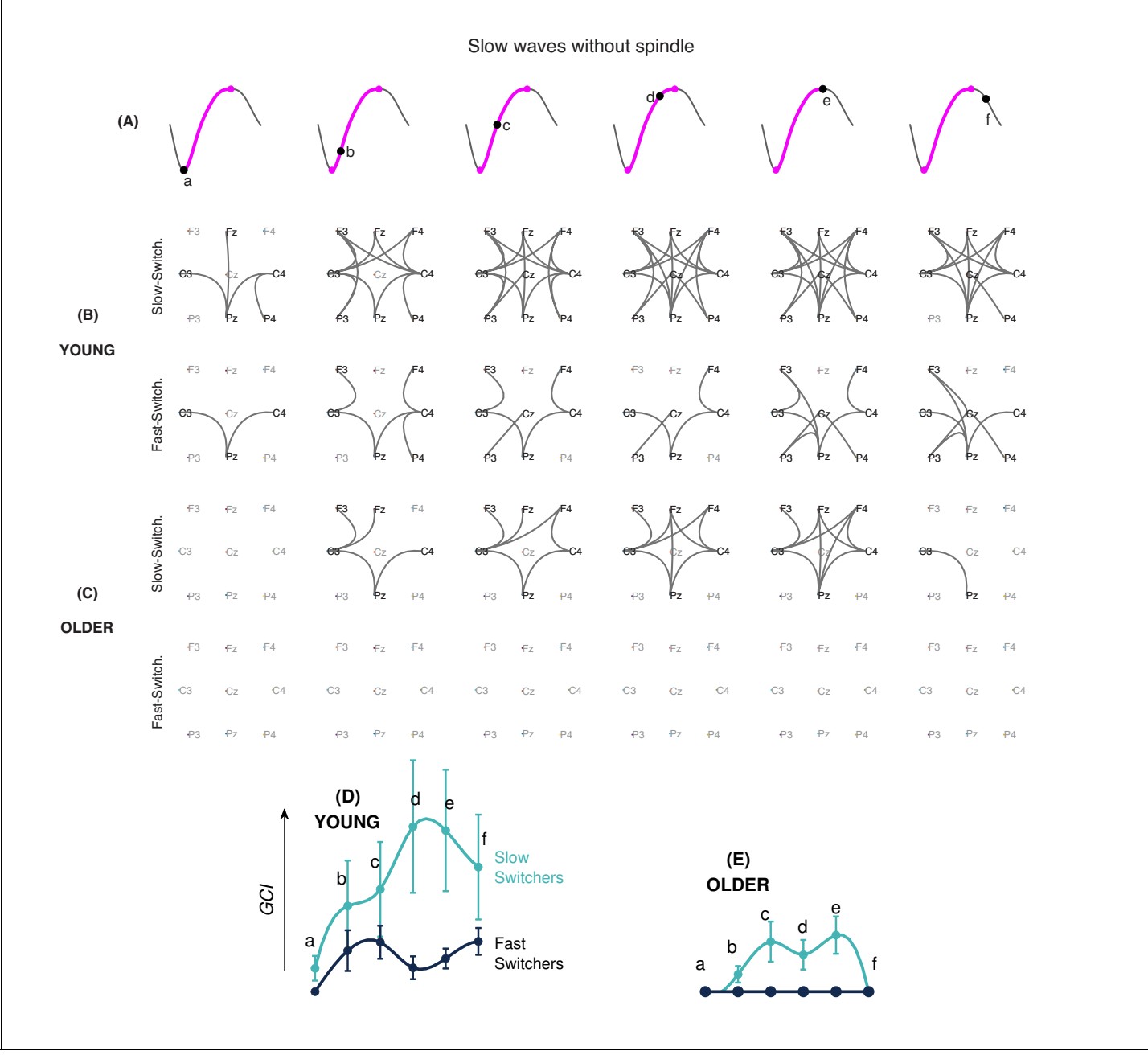

**Figure 5.** EEG connectivity signature of slow and fast switchers without sleep spindles in young and older individuals. (**A**) Illustration of the six different phases along the depolarization transition (a, b, c, d, and e) and the hyperpolarization transition of the slow wave (f). (**B and C**) EEG connectivity graphs using phase-locked connectivity (phase lag index, PLI) metrics and statistically assessed by non-parametric statistics for the slow (upper level) and fast (lower level) switchers without sleep spindles, in young (**B**) and older (**C**) individuals. (**D and E**) Global connectivity index (GCI) values at each phase of the slow wave, obtained by the summation of the PLI values across the significant electrode pairs. The GCI thus shows the weight of significant links obtained through non-parametric analyses. Slow switchers are represented in cyan, whereas the fast switchers are represented in dark blue. Graphs have a common scale and can therefore be compared.

individuals as compared to younger ones. Most importantly, when looking at homeostatic regulation, fast switchers showed a steeper decline between the sleep cycles across the night as compared to slow switchers. Using a data-driven approach, the results of this study thus distinguish two types of slow waves present in younger and older individuals, with specific characteristics that could embody complementary functional roles.

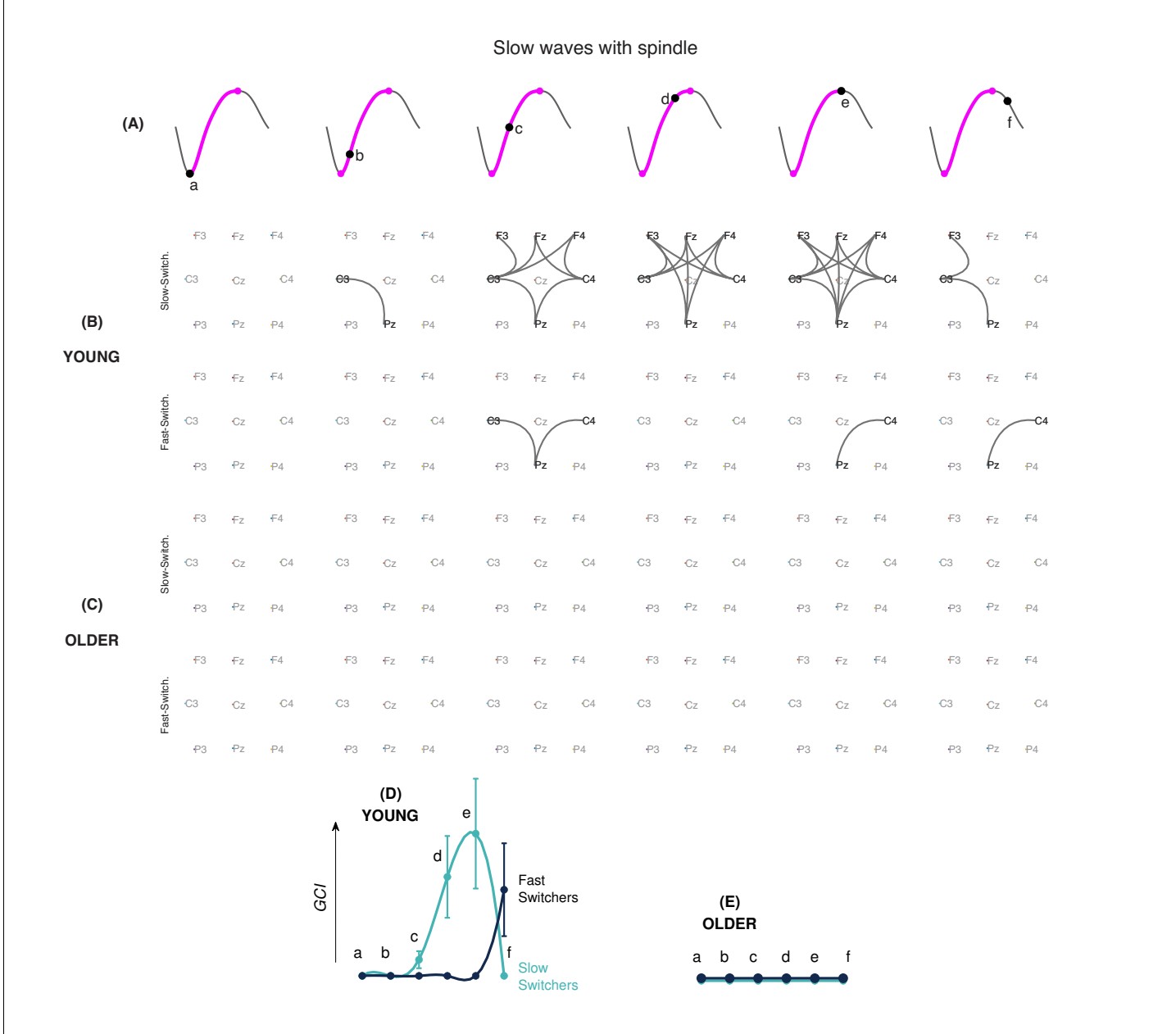

**Figure 6.** EEG connectivity signature of slow and fast switchers with sleep spindles in young and older individuals. (**A**) Illustration of the six different phases along the depolarization transition (a, b, c, d, and e) and the hyperpolarization transition of the slow wave (f). (**B and C**) EEG connectivity graphs, using phase-locked connectivity (phase lag index, PLI) metrics and statistically assessed by non-parametric statistics for the slow (upper level) and fast (lower level) switchers with sleep spindles, in young (**B**) and older (**C**) individuals. (**D and E**) Global connectivity index (GCI) values at each phase of the slow wave, obtained by the summation of the PLI values across the significant electrode pairs. Slow switchers are represented in cyan, whereas the fast switchers are represented in dark blue. Graphs have a common scale and can therefore be compared.

## Sleep slow waves are either slow or fast switchers

The frequencies related to half-wave components of filtered EEG were introduced decades ago in order to provide an alternative to the time-resolved spectral analysis of sleep (*Geering et al., 1993*). Such studies were mostly concerned with the half-waves defined by the zero-crossings of the entire high-pass-filtered EEG signals, although no consensus was reached. The present work introduces a new parameter in which half-waves and the associated frequency are defined from the depolarization transition of detected sleep slow waves. This intrinsic parameter, the *transition frequency*, objectively classifies sleep slow waves in humans into two categories: the slow and fast switchers. At the

physiological level, this frequency is associated with the synchronized depolarization of neurons. In the last year, *Nghiem et al., 2020* found two types of slow waves by analyzing temporal patterns of slow waves' down-states and up-states. While this distinction is seen in sleep, it does not seem to be present in anesthesia, which could point to a specific implication of one type of slow wave in sleep-dependent memory consolidation mechanisms specifically during sleep . Our results add crucial information to the recent published articles, showing that the two types of slow waves are under-pinned by a different EEG connectivity network dynamics across the depolarization transition itself and confirming the distinct homeostatic decline pattern. The use of a parameter more independent of the amplitude characteristic of the slow wave and associated specifically with the depolarization transition allows us to describe its intrinsic changes in aging. Taken altogether, these findings finally expand our understanding of the dichotomy described in the delta frequency of humans for years and how it is changing in the older population.

## Slow and fast switchers show distinct EEG connectivity dynamics

*Chauvette et al., 2010* described that in the cat's cortex, active states begin with the firing of a single neuron, leading to another neuron firing, then leading to a cascade of firing neurons. Here, we observed at a much larger scale that the EEG connectivity also increases as the depolarization occurs and could be linked to the higher communication and increased synaptic activity during the depolarization phase (*Chauvette et al., 2010*). In particular, the slow switchers in young adults involved a significant increase in EEG connectivity to reach a maximum at the depolarization state of the slow waves. A study by *Heib et al., 2013* showed that a longer duration of the depolarization phase of the slow wave was associated with better memory consolidation. Their hypothesis is that a longer depolarization phase could represent an increased possibility to effectuate an initial transfer of recent memory from the hippocampus to the cortex (*Heib et al., 2013*). Since our identified slow switchers show a slower transition frequency, their specific role in sleep-dependent cognitive processes should be investigated.

The increase in EEG connectivity at the scale of the oscillation seems to be complementary to the global disconnection we described in previous work in the delta band of N3 in the first sleep cycle of younger individuals (*Bouchard et al., 2019*). With the results of both studies, we can conjecture that this increase of connectivity during the depolarization transition of the slow switchers in young adults requires a global disconnection at a larger scale to make possible such a transient variation. The fast switchers, however, involved a lower connectivity index in young adults. Interestingly, if you look at *Figure 1B*, it seems that there exists a threshold (around 2.5 Hz) in the transition frequency, above which fast switchers are more difficult to produce. This critical frequency can be converted into a characteristic duration that corresponds to a period around 200 ms. Interestingly, this specific 200-ms duration was set up as the minimal time required for the establishment of connectivity networks, as recently measured by functional magnetic resonance imaging (fMRI) during resting states (*Baker et al., 2014*). We may hypothesize that if the connectivity of the slow wave is transiently associated with the establishment of a dynamic network, the depolarization of the slow wave can't be faster than the temporal scale needed for the setting of the network. Additionally, slow waves with a faster transition (the fast switchers with a frequency higher than 2.5 Hz) would not allow enough time for the dynamic network to take place.

It is also worth noting the variability of the connectivity index involved at each phase of the slow switchers' depolarization transition in younger individuals. This variability may reflect the diversity of networks recruited during such slow waves in those young adults. This idea of the transient reorganization of networks of a 'flexible brain' has been described in adults (*Baker et al., 2014*; *Spielberg et al., 2015*) and more recently in young children (*Yin et al., 2020*). Our results show that the flexibility of the slow switchers' connectivity, with or without spindles, is reduced in aging. Aging also significantly impacts the overall connectivity involved with the switchers. Although minimal connectivity persists for slow switchers in older adults, it was completely abolished for the fast switchers. This reduction in connectivity at the scale of the slow waves' depolarization transition in older individuals may be related to the general observation that the sleeping brain in aging remains functionally more connected at the scale of the sleep stages, namely N3 (*Bouchard et al., 2019*). It remains to be investigated if this change in connectivity dynamics could have precise functional consequences in aging, but the lack of EEG connectivity in older individuals suggests a decrease of flexibility in the ability to connect/disconnect and to mobilize the underlying network involved in slow waves. To

our knowledge, our study is the first to provide a functional connectivity analysis at different phases along the depolarization transition of the slow wave in humans and it is the first to describe its changes in the context of aging.

The presence of spindles over a slow wave modifies EEG connectivity in both slow and fast switchers. For instance, our results show that the EEG connectivity seems delayed when there is a concomitant spindle. For the younger adults, the EEG connectivity increase observed during the depolarization transition of the slow wave happened to be concomitant with the beginning of the spindle, whereas this connectivity had already risen in the absence of a spindle. This observation is not without recalling recent findings regarding the relationship between neural oscillations and the dynamics of functional connectivity (*Tewarie et al., 2019*): the spindling oscillation emerging on the top of the slow wave requires a 'static connectivity' from the latter (*Daffertshofer and van Wijk, 2011*). This is especially true for the most represented slow waves produced by adults, that is, the *fast switchers* of the young adults and the *slow switchers* of the older individuals. In aging, the presence of the spindle is associated with no changes in EEG connectivity as measured by the global connectivity index. A more exhaustive investigation of the dynamics of EEG connectivity in the interaction between slow waves and spindles connectivity networks in aging could likely contribute to better explaining the changes in sleep-dependent memory consolidation observed in the older population.

## Slow and fast switchers show distinct homeostatic responses

Our study shows that fast switchers undergo a steeper decline in the subsequent cycles, compared to slow switchers. When looking at the usual frequency of slow waves, *Hubbard et al., 2020* showed that prolonged waking periods are followed by a higher prevalence of faster waves at the beginning of the sleep period. They also showed that fast delta frequencies in mice and humans showed a steeper decline than slow delta frequencies after sleep deprivation. Other studies describing slow oscillations and delta waves using the usual frequency argued that low frequencies (<1 Hz) are less modulated by homeostatic pressure (*Achermann and Borbély, 1997*; *Campbell et al., 2006*). A recent study by *Kim et al., 2019* using a closed-loop optogenetic technique in rats was able to associate slow oscillations (<1 Hz) with consolidation of memory while slow waves (delta waves; <4 Hz) were involved in the forgetting process, showing dissociable and competing roles of the two rhythms in sleep-dependent memory consolidation. *Kim et al., 2019* also argued that the brain could accelerate the up-state transition of slow waves to better dissipate homeostatic pressure. Although it is unknown whether slow or fast switchers respond differently to a homeostatic challenge, we can hypothesize that fast switchers would be more involved in the response to a sleep challenge such as sleep deprivation.

## Slow and fast switchers evolve differently with aging

Older individuals in our study produced 60% of slow switchers compared to 40% for younger adults, which means that the prevalence of this type of oscillation significantly increases with advancing age. Compared to older participants, younger participants seem to have more efficient initiation and termination of slow waves' transition as they generate slow waves with a steeper slope (*Carrier et al., 2011*; *Ujma et al., 2019*). This rationale could partially explain the higher prevalence of slow switchers observed in our aging population, namely, that the latter might need an overall longer delay in polarity reversal. Also, age-related changes in homeostatic response could be responsible for the changes in slow wave production (*Tononi, 2009*). Indeed, older subjects show a significant decrease in their ability to increase the characteristics of slow waves (density, amplitude, slope, and duration) after a sleep deprivation, and these effects are more prominent in prefrontal and frontal derivations (*Lafortune et al., 2012*). One could hypothesize that these specific effects of aging on homeostasis response in frontal areas would reflect the underlying changes specifically in fast switchers as compared to slow switchers.

We showed that there is an age-related reduction in homeostatic response for both slow and fast switchers. However, the relative ratio of slow and fast switchers across all sleep cycles was maintained with age. While fast switchers were predominant in the first cycle, slow switchers predominated in all the other cycles for both young and old subjects. One could thus hypothesize that these oscillations have a different functional role to play across the night, both for young and older

subjects. Interestingly, the early night generally benefits verbal memory consolidation while subsequent sleep cycles could be more beneficial for procedural memory (*Plihal and Born, 1997*; *Gais and Born, 2004*). Future studies need to investigate the relative contribution of both fast and slow switchers to memory consolidation processes during sleep and overall sleep-dependent cognitive processes.

The animal literature on NREM slow wave parameters shows differences with humans. For instance, compared to humans, there is an age-related increase in frontal local field potentials (LFP) delta power (*Soltani et al., 2019*) and in slow waves' amplitude and slope in older mice, suggesting higher neuronal synchronization (*Panagiotou et al., 2017*; *McKillop et al., 2018*). Sleep deprivation protocols also showed higher sleep pressure (*Panagiotou et al., 2017*) and a similar sleep pressure discharge between young and older mice (*Wimmer et al., 2013*). While sleep researchers are trying to understand and explain the differences (*McKillop and Vyazovskiy, 2020*), the new parameter, that is, the transition frequency, brings a new angle of analysis and could lead to interesting insights into this problem, for example, by looking at the proportion of slow and fast switchers, their proportion in sleep deprivation protocols, and their pattern of homeostatic decline and brain functional connectivity. While more sleep deprivation studies are needed to understand the functional role of slow and fast switchers and their value for the aging brain, looking into slow and fast switchers in animals would enhance our understanding of the sleeping brain.

## Conclusion

This study is the first to use the *transition frequency* of slow waves to introduce and to study the slow and fast switchers in the slow wave spectrum, which were identified in both young and older adults. Slow and fast switchers present different connectivity dynamics along their depolarization transition, with slow switchers having a higher connectivity than fast switchers. They are also differently modulated during the night, with fast switchers showing steeper decreases at the beginning of the night. Aging was associated with a higher number of slow switchers than fast switchers, an overall lower EEG connectivity across the depolarization transition of slow waves, and a flatter homeostatic decline of both slow wave types across the night. Those results regarding slow waves likely imply different functional mechanisms associated with slow and fast switchers that could be modified in aging.

## Acknowledgements

The authors would like to thank Sonia Frenette for her help with data collection and analysis and Carrie Schipper, for reviewing the manuscript. This work was supported by the Canadian Institutes of Health Research (CIHR), grant number 190750 (JC), the NSERC-Discovery programs (J-ML and JC), the CIHR Vanier scholarship (MB), and a CIHR postdoctoral award (P-OG).

All codes (*Lina, 2021a*) and transformed data used for all the analyses and most specifically to produce all of the figures of this article can be freely accessible using this link: https://github.com/jmlina/Slow_Wave_Switchers, (copy archived at swh:1:rev:3af1be579bd6d5aac1718d49c85e8af9c17541c1), *Lina, 2021b*.

## Additional information

### Funding

| Funder | Grant reference number | Author |
| --- | --- | --- |
| Canadian Institutes of Health Research | Vanier scholarship | Maude Bouchard |
| Canadian Institutes of Health Research | 190750 | Julie Carrier |
| Natural Sciences and Engineering Research Council of Canada | NSERC-Discovery Program | Jean-Marc Lina<br>Julie Carrier |
| Canadian Institutes of Health Research | Postdoctoral Fellowship | Pierre-Olivier Gaudreault |

The funders had no role in study design, data collection and interpretation, or the decision to submit the work for publication.

## Author contributions
Maude Bouchard, Conceptualization, Investigation, Methodology, Writing - original draft, Writing - review and editing; Jean-Marc Lina, Conceptualization, Formal analysis, Supervision, Writing - review and editing; Pierre-Olivier Gaudreault, Formal analysis, Writing - review and editing; Alexandre Lafrenière, Jonathan Dubé, Nadia Gosselin, Writing - review and editing; Julie Carrier, Resources, Supervision, Funding acquisition, Writing - review and editing

## Author ORCIDs
Maude Bouchard ⬚ https://orcid.org/0000-0002-7858-4136
Pierre-Olivier Gaudreault ⬚ https://orcid.org/0000-0002-9863-4436
Julie Carrier ⬚ https://orcid.org/0000-0001-5311-2370

## Ethics
Human subjects: The protocol was approved by the ethics committee of the Hôpital du Sacré-Coeur de Montréal and performed in accordance with the relevant guidelines and regulations. Participants provided informed consent and received financial compensation for their participation. (CMER-RNQ 08-136 08-002).

## Decision letter and Author response
Decision letter https://doi.org/10.7554/eLife.64337.sa1
Author response https://doi.org/10.7554/eLife.64337.sa2

# Additional files

## Supplementary files
- Supplementary file 1. Demographic and polysomnographic variables for young and older subjects.
- Transparent reporting form

## Data availability
All codes and transformed data used for all the analyses and most specifically to produce all of the figures of the paper can be freely accessible using this link : https://github.com/jmlina/Slow_Wave_Switchers (copy archived at https://archive.softwareheritage.org/swh:1:rev:3af1be579bd6-d5aac1718d49c85e8af9c17541c1). Dataset cannot be shared as participants did not give consent for data sharing. For the raw data, a request needs to be formulated to the ethic committee of the Hôpital de Sacré-Coeur de Montréal, as rawdata of human participants cannot be made public under Québec's law. The data provided will be anonymized. Researchers who request access to the data will need to provide their research protocol and their IRB approval for this protocol. The documents will be studied by the owner of the database (Julie Carrier) who will then also submit to her institution's REB for authorization to share the data. Data requests should be addressed to: Julie Carrier (PI): julie.carrier.1@umontreal.ca Sonia Frenette (in cc) : sonia.frenette@umontreal.ca.

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
