## [Decision Letter]

**Acceptance summary:**

The authors characterize a specific novel electrophysiological property of slow waves during non-rapid eye movement (NREM ) sleep in humans related to the speed of transition (termed "transition frequency") between the two defining hyperpolarized and depolarized brain states associated with NREM slow waves. This study provides important new data to better understand the network dynamics giving rise to slow waves, the biological role of slow waves, how their expression changes with age, and offers critical implications for the biology of sleep-dependent cognition and cognitive impairment in later life.

**Decision letter after peer review:**

Thank you for submitting your article "Sleeping at the Switch" for consideration by *eLife*. Your article has been reviewed by 3 peer reviewers, and the evaluation has been overseen by a Reviewing Editor and Chris Baker as the Senior Editor. The following individual involved in review of your submission has agreed to reveal their identity: Ksenia V. Kastanenka (Reviewer #3).

Essential revisions:

1) The manuscript findings need to be more fully embedded into the existing literature, particularly with respect to non-human animal findings characterizing the physiological properties of slow waves and their distinctions along different dimensions, and further clarity regarding the novelty of this metric in relation to this literature is warranted.

2) The analysis methods implemented require greater detail and clarity.

3) Clear statistical demonstration of the independence of the transition frequency from slow wave amplitude and slope metrics is absent and was noted by multiple reviewers to be a critical concern.

*Reviewer #1:*

In this work, the authors propose a functional subdivision of EEG slow waves in humans based on "transition frequency" between putative UP and DOWN states. The data suggest an existence of two distinct subtypes of slow waves, called slow switchers and fast switchers. It appears that these two types differ with respect to the response to homeostatic sleep pressure, with respect to ageing and in relation to "connectivity". Overall, this is an interesting study, that provides potentially important new data that can help to understand the biological role of sleep slow waves.

I have specific comments as follows:

1. I would not assume that the polarity of slow waves recorded using scalp EEG can be easily related to neuronal UP and DOWN states or states of membrane depolarisation and hyperpolarisation. I urge the authors to discuss this more critically and amend terminology where relevant because this is misleading.

2. There is substantial animal literature, which investigated the relationship between EEG, LFP and neuronal population ON and OFF periods or UP/DOWN states. This earlier work in mice and rats addressed the effects of sleep-wake history and ageing on cortical slow waves and their neuronal correlates, including slow wave characteristics, such as slopes and the underlying neuronal synchrony; however, these studies are not mentioned.

3. What is the difference between the new metric proposed here and previously used slow wave slope, which was linked to synchronisation at the ON-OFF and OFF-ON transitions? The authors argue that their approach is based on "a parameter free of the amplitude characteristic of the slow wave", but this has not been demonstrated. It is likely that transition frequency is related to slow wave amplitude, even if the latter is not explicitly included in the calculation. Please provide further analyses to evaluate how transition frequency relates to slow wave amplitude, duration, and slopes to enable comparisons with earlier studies.

4. Figure 1 illustrates how the cut-off of 1.2 Hz was obtained from the data, but essential details are lacking. Do I understand correctly that the scatter plot shows all individual slow waves pooled across all subjects? Could you show individual subjects separately? Do you still see bimodality at the individual subject level? Does the difference between younger and older groups still manifest when you plot distributions of transition frequencies for individual younger and older subjects, and can this be tested statistically? I wondered if you were to replicate the analyses used by Hubbard et al., 2020 on your data, can you see bimodality they report in mice?

5. It is stated in Methods: "All slow waves were equally time referenced by choosing the zero phase at the maximum of the depolarization." It is not clear what is "maximum" here? I presume it is not uncommon that there are multiple positive peaks on the "depolarisation" part of the slow wave, which complicates this analysis. Was the number of peaks within a slow wave different between older and younger subjects? I wondered if the two categories of slow waves correspond to those with one peak and two peaks riding on the top of the "depolarisation/up" phase?

6. I wondered if the "critical frequency" or the dynamics of "fast" and "slow" switchers would change (or an additional category emerge) if you filter the signals at a higher frequency, such as at 6 Hz, rather than 4 Hz (e.g. see https://pubmed.ncbi.nlm.nih.gov/10607082/)

7. The analysis of EEG connectivity is interesting, but I wondered if there is a relationship between occurrence of multiple peaks on the depolarisation phase and connectivity? Riedner et al. study suggested that multipeak slow waves can reflect "collision" of slow waves originating from distant cortical areas: https://pubmed.ncbi.nlm.nih.gov/18246974/

Generally it is a work conducted up to a high standard, but the main issue that remains to be addressed is how robust the approach used here to changes in specific parameters, such as how the signals were filtered. I would also recommend providing some further analyses to assess how transition frequency relates to slow wave amplitude and slope.*Reviewer #2:*

Here, the authors present data regarding the occurrence of (relatively speaking) slow and fast onset slow waves in younger and older adults, and some phenomenology surrounding their occurrence. There is a reasonable argument to be made that this may represent an actual dichotomy – e.g. bimodal distribution of onset time (essentially the slope). The authors claim that this is a feature independent of wave amplitude, but no data is shown to this effect. This makes the interpretation of the significance of these data problematic, as many of the features about "fast switcher" waves are already well known regarding high-amplitude waves. Thus it is hard to know what is really new here – as much is already known about changes in amplitude with aging.

The authors have developed some new metrics to characterize features of slow wave waveforms during NREM sleep. They propose that there are two classes of slow waves and show some suggestive data that is indeed the case. For example, they show a clear bimodal distribution of transition frequencies (seemingly a measure of transition slope). It is not clear how this relates to wave amplitude- data addressing this are not shown – but it stands to reason that this feature maps onto wave amplitude very well. Because much is already known about amplitude of waves' relationship to synchrony across recording sites, and aging, and occurrence across the night, it is very unclear if this is just a retelling of the same story using a different metric.

An analysis of how the new metric relates to waves' amplitudes is absolutely necessary for the readers to know if there is anything clearly different between this study and what is already known.

A fair bit of the methodology around analyses data is not present, making it questionable what different figures are really showing – for example, it is unclear which stages of NREM sleep are used for various measures, and if these waves' occurrence differs between those stages.

*Reviewer #3:*

In this manuscript the authors aim to distinguish between two types of slow waves based on a new parameter characterizing the time delay from the up-to-down state, the transition frequency. Sleep and slow waves were monitored in young and older adults in good physical and mental health. The authors report distinction between two types of slow waves, slow switchers and fast switchers, which were detected in both age groups with the older individuals showing a higher proportion of slow switchers.

Strengths:

The manuscript is well-written and question under investigation is well-articulated.

The work is novel and significant to the field.

Conclusions are supported by the data.

Weaknesses: None

[Editors' note: further revisions were suggested prior to acceptance, as described below.]

Thank you for resubmitting your work entitled "Sleeping at the Switch" for further consideration by *eLife*. Your revised article has been evaluated by Chris Baker (Senior Editor) and a Reviewing Editor.

The manuscript has been improved but there was one remaining issue that needs to be addressed, as outlined below:

While the reviewers agreed that the manuscript was substantially improved through revisions, the consensus was that further consideration in the discussion of the study findings was needed in the context of the publication by Geering et al. (https://pubmed.ncbi.nlm.nih.gov/10607082/), which reports the differential temporal dynamics of slow waves at different frequencies and the impact of analysis methods on their metrics.

---

## [Author Response]

Essential revisions:1) The manuscript findings need to be more fully embedded into the existing literature, particularly with respect to non-human animal findings characterizing the physiological properties of slow waves and their distinctions along different dimensions, and further clarity regarding the novelty of this metric in relation to this literature is warranted.2) The analysis methods implemented require greater detail and clarity.3) Clear statistical demonstration of the independence of the transition frequency from slow wave amplitude and slope metrics is absent and was noted by multiple reviewers to be a critical concern.Reviewer #1:In this work, the authors propose a functional subdivision of EEG slow waves in humans based on "transition frequency" between putative UP and DOWN states. The data suggest an existence of two distinct subtypes of slow waves, called slow switchers and fast switchers. It appears that these two types differ with respect to the response to homeostatic sleep pressure, with respect to ageing and in relation to "connectivity". Overall, this is an interesting study, that provides potentially important new data that can help to understand the biological role of sleep slow waves.I have specific comments as follows:1. I would not assume that the polarity of slow waves recorded using scalp EEG can be easily related to neuronal UP and DOWN states or states of membrane depolarisation and hyperpolarisation. I urge the authors to discuss this more criticall, and amend terminology where relevant because this is misleading.

We agree with the reviewer and we clarified different sections in the article.

1) Modification in the Introduction section of the main manuscript.

1. “More recently, animal and human studies brought to light new evidence of two types of slow waves based on the up and down state duration: one showing positive correlation between consecutive up and down state duration, and another one showing negatively correlated up and down state durations.”

Was replaced by

“More recently, animal and human studies brought to light new evidence of two types of slow waves based on the positive and negative state duration: one showing positive correlation and another one showing a negative correlation between the two.” (p.3, lines 23-25)

2. “Here, we propose to describe the dichotomy in the δ frequency range based on a new parameter characterizing the time delay from the up-to-down state: the *transition frequency*.”

was replaced by

“Here, we propose to describe the dichotomy in the δ frequency range based on a new parameter characterizing the time delay from the maximum negative point of the EEG slow wave to the maximum positive point of the slow wave: the *transition frequency*.” (page 4, lines 2-3)

3. “Therefore, our ability to disentangle the influence of slow wave amplitude over our typical metric of the down-to-up state transition, e.g., the slope, is compromised in aging.”

Was replaced by

Therefore, our ability to disentangle the influence of slow wave amplitude over the slope, is compromised in aging. (page 5, lines 2-3)

4. “If τ denotes the delay of the down-to-up-state transition (see Figure 1A)”.

was replaced by

“If τ denotes the delay of the transition from the maximum negative point to the maximum positive point of the slow wave”. (page 8, lines 22-23)

2) Modification in the Discussion section of the main manuscript.

1. “of their *transition frequency* (down-to-up state transition)”

was replaced by

“(the transition between the maximum negative point and the maximum positive point of the slow wave)”. (page 19, lines 8-9)

2. “is the *transition frequency* which is strictly associated with the half-waves from the down-to-up state transition”.

was replaced by

“is the *transition frequency* which is strictly associated with the half-waves related to the depolarization transition of the slow wave.” (page 19, lines 23-24)

3. “Compared to older participants, younger participants seem to have more efficient initiation and termination of slow waves down-state and up-state as they are generating slow waves with a steeper slope”

was replaced by

“Compared to older participants, younger participants seem to have more efficient initiation and termination of slow waves’ transitions as they are generating slow waves with a steeper slope”. (page 23, lines 23-24-25)

2. There is substantial animal literature, which investigated the relationship between EEG, LFP and neuronal population ON and OFF periods or UP/DOWN states. This earlier work in mice and rats addressed the effects of sleep-wake history and ageing on cortical slow waves and their neuronal correlates, including slow wave characteristics, such as slopes and the underlying neuronal synchrony; however, these studies are not mentioned.

We thank the reviewer for his/her suggestion and modifications to the Discussion section of the main manuscript were made to include information about animal literature.

Changes made to the Introduction section of the main manuscript:

“The slope of the slow wave (the rate of amplitude change from the negative to the positive peak) is generally described as the best measure to assess synaptic strength and sleep homeostasis compared to other classic parameters”.

was replaced by

“The slope of the slow wave (the rate of amplitude change from the negative to the positive peak) is associated with the recruitment/decruitment of the neuronal population with a steeper slope showing a quicker recruitment (Vyazovskiy et al., 2011).” (p.4, line 12-15)

Added references:

Vyazovskiy, V. V., Cirelli, C., and Tononi, G. (2011). Electrophysiological correlates of sleep homeostasis in freely behaving rats. *Progress in brain research*, *193*, 17–38.

Changes made to the Discussion section of the main manuscript:

“The animal literature on NREM slow waves parameters shows differences with humans. For instance, compared to humans, there is an age-related increase in frontal LFP δ power (Soltani et al., 2019) and in slow waves’ amplitude and slope in older mice suggesting higher neuronal synchronization (Panagiotou et al., 2017; McKillop et al., 2018). Sleep deprivation protocols also showed higher sleep pressure (Panagiotou et al., 2017) and similar sleep pressure discharge between young and older mice (Wimmer et al., 2013). While sleep researchers are trying to understand and explain the differences (McKillop et al., 2020), the new parameter i.e., transition frequency brings a new angle of analysis and could lead to interesting insight into this problem, for example, by looking at the proportion of slow and fast switchers, their proportion in sleep deprivation protocols and their pattern of homeostatic decline and brain functional connectivity. While more sleep deprivation studies are needed to understand the functional role of slow and fast switchers and their value for the aging brain, looking into slow and fast switchers in animals would enhance in our understanding of the sleeping brain.” (page 24, line 21-25 + p. 25, line 1-8)

Added references:

McKillop LE, Fisher SP, Cui N, Peirson SN, Foster RG, Wafford KA, Vyazovskiy VV.

(2018). Effects of Aging on Cortical Neural Dynamics and Local Sleep Homeostasis in Mice. *J Neurosci.* 18;38(16):3911-3928.

Panagiotou, M., Vyazovskiy, V.V., Meijer, J.H. and Deboer T. (2017) Differences in

electroencephalographic non-rapid-eye movement sleep slow-wave characteristics between young and old mice. *Sci Rep.* 3;7:43656.

Soltani, S., Chauvette, S., Bukhtiyarova, O., Lina, J.M., Dubé, J., Seigneur, J., Carrier, J.

and Timofeev, I. (2019). Sleep-Wake Cycle in Young an Older Mice. *Front. Syst. Neurosci*. 13(51).

Wimmer ME, Rising J, Galante RJ, Wyner A, Pack AI, et al. (2013). Aging in Mice

Reduces the Ability to Sustain Sleep/Wake States. *PLOS ONE* 8(12): e81880.

3. What is the difference between the new metric proposed here and previously used slow wave slope, which was linked to synchronisation at the ON-OFF and OFF-ON transitions? The authors argue that their approach is based on "a parameter free of the amplitude characteristic of the slow wave", but this has not been demonstrated. It is likely that transition frequency is related to slow wave amplitude, even if the latter is not explicitly included in the calculation. Please provide further analyses to evaluate how transition frequency relates to slow wave amplitude, duration and slopes to enable comparisons with earlier studies.

We thank the reviewer for giving us the opportunity to clarify this important point. The next figure and the following comments explain in more detail how the transition frequency is a slow wave intrinsic feature, giving a unique and non-arbitrary way of classifying slow waves. It also shows that the classification is mainly independent from its slope and amplitude.

Consider Figure 2—figure supplement 1:

We considered all the detected slow waves in the younger and older cohorts. The reasoning is explained with the older cohort, though it can be done with the younger individuals as well. The panel (A) shows the distribution of the transition frequency defined as slope(2×amp), slope and amp being the slope and the peak-to-peak amplitude respectively. As shown in the manuscript, this distribution can be accurately fitted with a mixture of two Gaussians: one Gaussian with a mean frequency of 0.93 Hz and the other with a mean frequency of 1.63 Hz. The dashed line in (A) is the crossing of the two Gaussians of the mixture, at 1.2 Hz and defines the threshold that allows separating the slow switchers from the fast switchers.

The graphics in panels (B) and (C) display the distribution of the slopes and the amplitudes. Both distributions are well fitted with a γ and an exponential distribution respectively, as shown with thicker lines. It is worth noting that the distributions of the slope and the amplitudes reflect a unique process whereas the bimodal mixture of the transition frequency distribution reveals the underlying distinction of two types of slow waves.

In order to complement this description, we provide the scatter plot of the panels (D) and (E) that displays the slow switchers (dark grey) and fast switchers (light grey) with respect to the slope or the amplitude and the transition frequency. In (E), the SWs scatter inside a cone defined by the lower and higher peak to peak amplitudes. Although the lower side of the cone is related to the minimal value of the peak-to-peak amplitude as defined in the detector (75μV), the upper side of the cone indicates that the amplitude peak to peak is bounded and cannot exceed ∼433μV in the older individuals (this value is ∼515μV for the young). It is also worth noting that SWs with slope < 180 μV/sec are essentially slow switchers. However, as the SWs’ slope increases, we obtain a mixture of slow and fast switchers with a growing proportion of fast switchers. For a given slope above this threshold (180 μV/sec), we can find both slow and fast switchers. There are a few slow waves with a very large slope (>∼834 μV/sec) that are fast switchers. We interpret that in some circumstances, slow waves with a small or very large slope could have been singularly categorized. Nevertheless, an unambiguous classification of the slow waves can be done only from the transition frequency. Finally, the panel (F) displays the scatter plot of the slow waves with respect to the amplitude and the transition frequency. No correlation can relate those two features of the slow waves.

Changes made to the Introduction section of the main manuscript:

“A novel metric that captures the transition speed without being affected by amplitude needs to be developed.”

Was replaced by:

“A novel metric that captures the transition speed and that is more independent of amplitude needs to be developed.” (p.4, line 20-21)

Changes made to the Discussion section of the main manuscript:

“The use of a parameter free of the amplitude characteristic of the slow wave and associated specifically with the depolarization transition allows us to describe its intrinsic changes in aging.”

Was replaced by

“The use of a parameter more independent of the amplitude characteristic of the slow wave and associated specifically with the depolarization transition allows us to describe its intrinsic changes in aging.” (p.20, line 7-8)

4. Figure 1 illustrates how the cut-off of 1.2 Hz was obtained from the data, but essential details are lacking. Do I understand correctly that the scatter plot shows all individual slow waves pooled across all subjects? Could you show individual subjects separately? Do you still see bimodality at the individual subject level? Does the difference between younger and older groups still manifest when you plot distributions of transition frequencies for individual younger and older subjects, and can this be tested statistically? I wondered if you were to replicate the analyses used by Hubbard et al., 2020 on your data, can you see bimodality they report in mice?

The scatter plot indeed shows all individual slow waves pooled across all subjects (young and older adults). However, in the Results section, the Figure 3(B) and (C) show the Gaussian distributions of the mixture for each subject separately (see legend of figure 3: (B) and (C) The distribution of probabilities of slow waves being slow (cyan) or fast switchers (dark blue) in younger and older individuals, respectively, with each curve representing one participant.). The figure therefore shows this bimodality occurring at the individual level as well as for younger and older adults independently. Now, Figure 3—figure supplement 2 shows the same bimodality when we look at slow waves with and without concomitant sleep spindles.

In Hubbard et al. (2020), the authors describe two sets of slow waves defined when comparing sleep-deprived mice to control ones and using different parameters than the transition frequency. Although it is not possible to look at sleep deprivation in our current data as none of the participants underwent a sleep deprivation protocol, we do see one type of slow wave being more prevalent than the other at the beginning of the night (the fast switchers) when the homeostatic pressure is at its highest (See Figure 3). The fact that Hubbard et al. (2020) show that slow waves “accelerate” after a prolonged waking period tends to corroborate our findings. Figure 1 in our paper shows that we couldn’t have defined slow and fast switcher based on the frequency of the slow waves and Figure 2—figure supplement 1 shows that we couldn’t have used the slow waves’ slope nor the amplitude.The legend of Figure 3 was clarified to emphasize the mixture of Gaussians present at the individual level as displayed in Figure 3(B) and (C).

Changes made to the Discussion of the manuscript:

Addition: “When looking at the usual frequency of slow waves, Hubbard et al. (2020) show that prolonged waking periods are followed by a higher prevalence of faster waves at the beginning of the sleep period” (p.23, lines 5-7).

5. It is stated in Methods: "All slow waves were equally time referenced by choosing the zero phase at the maximum of the depolarization." It is not clear what is "maximum" here? I presume it is not uncommon that there are multiple positive peaks on the "depolarisation" part of the slow wave, which complicates this analysis. Was the number of peaks within a slow wave different between older and younger subjects? I wondered if the two categories of slow waves correspond to those with one peak and two peaks riding on the top of the "depolarisation/up" phase?

The maximum of the depolarization phase is illustrated in Figure 1A and was defined according to our detection criteria (i.e., defined as the absolute maximum of amplitude). The same points are usually used to define the slope and amplitude of slow waves in human studies. We therefore did not identify a multiple local maximum within the slow wave that could be interpreted in various ways, including the possible ringing effect of the filter. Our analyses considered the unambiguous global maximal peak for each phase of the slow wave. Moreover, visual inspection of the slow wave shows marginal occurrence of other local maximum peaks on the filtered signal. Although small wavelets might be occasionally identified in the main slow wave phenomenon when using surface EEG in humans, it can be very difficult to interpret the underlying neurophysiology because of the spatial resolution (the dense coverage of the HD-EEG may help to reproduce the local peaks across neighboring derivations).

6. I wondered if the "critical frequency" or the dynamics of "fast" and "slow" switchers would change (or an additional category emerge) if you filter the signals at a higher frequency, such as at 6 Hz, rather than 4 Hz (e.g. see https://pubmed.ncbi.nlm.nih.gov/10607082/)

This question raised by the reviewer is indeed interesting, but we didn’t analyze higher frequencies as our objective was to investigate slow waves in the usual frequency range as defined in the human literature. Even if we were to have used a 6Hz filter, other parameters used to identify SW (negative peak below -40 µV, a peak-to-peak amplitude above 75 uV, the duration of negative deflection between 1500 and 125 ms, and the duration of positive deflection not exceeding 1000 ms) would limit the number of waves identified between 4Hz and 6Hz and would thus unlikely move the transition frequency delimiting slow and fast switchers. Moreover, as seen in Figure 1 in the manuscript, the transition frequency is upper bounded and an investigation at higher frequency would likely not exhibit an other class of switchers.

7. The analysis of EEG connectivity is interesting, but I wondered if there is a relationship between occurrence of multiple peaks on the depolarisation phase and connectivity? Riedner et al. study suggested that multipeak slow waves can reflect "collision" of slow waves originating from distant cortical areas: https://pubmed.ncbi.nlm.nih.gov/18246974/

As mentioned previously, the occurrence of multiple local maxima on the filtered slow wave was marginal and difficult to interpret. The increase in functional connectivity happening in young adults during the depolarization phase of the slow wave is at its highest at the positive peak. Being able to identify “colliding” slow waves from distant cortical areas would definitely be interesting but is unfortunately out of the scope of this manuscript, especially with the EEG montage we used. This would be easier to assess with a high-density EEG that may be able to capture such interference with a higher spatial resolution.

Reviewer #2:Here, the authors present data regarding the occurrence of (relatively speaking) slow and fast onset slow waves in younger and older adults, and some phenomenology surrounding their occurrence. There is a reasonable argument to be made that this may represent an actual dichotomy – e.g. bimodal distribution of onset time (essentially the slope). The authors claim that this is a feature independent of wave amplitude, but no data is shown to this effect. This makes the interpretation of the significance of these data problematic, as many of the features about "fast switcher" waves are already well known regarding high-amplitude waves. Thus it is hard to know what is really new here – as much is already known about changes in amplitude with aging.The authors have developed some new metrics to characterize features of slow wave waveforms during NREM sleep. They propose that there are two classes of slow waves, and show some suggestive data that is indeed the case. For example, they show a clear bimodal distribution of transition frequencies (seemingly a measure of transition slope). It is not clear how this relates to wave amplitude- data addressing this are not shown – but it stands to reason that this feature maps onto wave amplitude very well. Because much is already known about amplitude of waves' relationship to synchrony across recording sites, and aging, and occurrence across the night, it is very unclear if this is just a retelling of the same story using a different metric.An analysis of how the new metric relates to waves' amplitudes is absolutely necessary for the readers to know if there is anything clearly different between this study and what is already known.

We thank the reviewer for giving us the opportunity to clarify this important point. Figure 2—figure supplement 1 and the following comments explain in more detail how the transition frequency is a slow wave intrinsic feature, giving a unique and non-arbitrary way of classifying slow waves. It also shows the classification is mainly independent from its slope and amplitude.

We considered all the detected slow waves in the younger and older cohorts. The reasoning is explained with the older cohort, though it can be done with the younger individuals as well. The panel (A) shows the distribution of the transition frequency defined as slope(2×amp), slope and amp being the slope and the peak-to-peak amplitude respectively. As shown in the manuscript, this distribution can be accurately fitted with a mixture of two Gaussians: one Gaussian with a mean frequency of 0.93 Hz and the other with a mean frequency of 1.63 Hz. The dashed line in (A) is the crossing of the two Gaussians of the mixture, at 1.2 Hz and defines the threshold that allows separating the slow switchers from the fast switchers.

The graphics in panels (B) and (C) display the distribution of the slopes and the amplitudes. Both distributions are well fitted with a gamma and an exponential distribution respectively, as shown with thicker lines. It is worth noting that the distributions of the slope and the amplitudes reflect a unique process whereas the bimodal mixture of the transition frequency distribution reveals the underlying distinction of two types of slow waves.

In order to complement this description, we provide the scatter plot of the panels (D) and (E) that displays the slow switchers (dark grey) and fast switchers (light grey) with respect to the slope or the amplitude and the transition frequency. In (E), the SWs scatter inside a cone defined by the lower and higher peak to peak amplitudes. Although the lower side of the cone is related to the minimal value of the peak-to-peak amplitude as defined in the detector (75μV), the upper side of the cone indicates that the amplitude peak to peak is bounded and cannot exceed ∼433μV in the older individuals (this value is ∼515μV for the young). It is also worth noting that SWs with slope < 180 μV/sec are essentially slow switchers. However, as the SWs’ slope increases, we obtain a mixture of slow and fast switchers with a growing proportion of fast switchers. For a given slope above this threshold (180 μV/sec), we can find both slow and fast switchers. There are a few slow waves with a very large slope (>∼834 μV/sec) that are fast switchers. We interpret that in some circumstances, slow waves with a small or very large slope could have been singularly categorized. Nevertheless, an unambiguous classification of the slow waves can be done only from the transition frequency. Finally, the panel (F) displays the scatter plot of the slow waves with respect to the amplitude and the transition frequency. No correlation can relate those two features of the slow waves.

A fair bit of the methodology around analyses data is not present, making it questionable what different figures are really showing – for example, it is unclear which stages of NREM sleep are used for various measures, and if these waves' occurrence differs between those stages.

Slow waves and sleep spindles were detected on both N2 and N3 sleep stages. Both stages were thus used in our measures in order to include all slow waves and spindles.

See “Slow waves were detected automatically on artifact-free NREM (N2 and N3) epochs on all electrodes using previously published criteria” (page 7, line 20-21)

and

“Spindles were automatically detected on artifact-free NREM (N2 and N3) epochs on all electrodes using a previously published algorithm” (page 8, line 4-5)

The information about NREM sleep stages used was added to the legend of Figure 1, 2, and 3 of the main manuscript.

[Editors' note: further revisions were suggested prior to acceptance, as described below.]

The manuscript has been improved but there was one remaining issue that needs to be addressed, as outlined below:While the reviewers agreed that the manuscript was substantially improved through revisions, the consensus was that further consideration in the discussion of the study findings was needed in the context of the publication by Geering et al. (https://pubmed.ncbi.nlm.nih.gov/10607082/), which reports the differential temporal dynamics of slow waves at different frequencies and the impact of analysis methods on their metrics.

We want to thank the reviewers and editors for pointing us this article. The paper is interesting as it introduces the idea that half waves can bring complementary information to spectral analyses. We recognize there is similarities between this work and ours. However, there are also important methodological differences which prevent our results to be contaminated by some bias noted in the article.

The internal frequency we discuss in our paper is based on a different rationale than what proposed by Geering et al. We studied the frequency of the depolarization transition of detected sleep slow waves, whereas they analyzed the entire high passed EEG signal and they focused on half waves obtained from zero-crossing. Using zero crossings can introduce a bias as it can be arbitrarily manipulated through the choice of high pass filter, a notion the authors acknowledged themselves at the end of their discussion. It is worth noting that the authors also mentioned the ‘through to peak’ transition still using zero-crossing points but on the first order derivative of the signal. This is more in the spirit of our work. However, in our study, we only considered the detected slow waves from which we extracted ‘through to peak’ transition directly. Despite some correspondence in the kind of local characteristics, the two approaches are very different in terms of the signals under consideration. We argue that using the depolarization transition with the max and min peak of detected slow-waves allow us to evaluate a more physiological process, free of the methodological bias discussed in Geering et al., paper.

As suggested by the reviewer, we added a section at the beginning of the discussion to acknowledge their scientific contribution.

This paragraph was added to the Discussion section of the main manuscript.

The frequencies related to half-wave components of filtered EEG were introduced decades ago in order to provide an alternative to the time-resolved spectral analysis of sleep (Geering et al., 1993).

Such studies were mostly concerned with the half-waves defined by the zero-crossings of the entire high pass filtered EEG signals, although no consensus was reached. The present work introduces a new parameter in which half-waves and the associated frequency are defined from the depolarization transition of detected sleep slow waves. This intrinsic parameter, the *transition frequency,* objectively classifies sleep slow waves in humans into two categories: the slow and fast switchers. (p.15, Lines: 2-7).